# Decoding reveals the neural representation of perceived and imagined musical sounds

**David R. Quiroga-Martinez**[1,2*], **Gemma Fernández Rubio**[3], **Leonardo Bonetti**[3,4,5], **Kriti G. Achyutuni**[1], **Athina Tzovara**[1,6,7], **Robert T. Knight**[1], **Peter Vuust**[3]

**1** Helen Wills Neuroscience Institute & Department of Psychology and Neuroscience, University of California Berkeley, Berkeley, California, United States of America, **2** Psychology Department, University of Copenhagen, Copenhagen, Denmark, **3** Center for Music in the Brain, Department of Clinical Medicine, Aarhus University and The Royal Academy of Music, Aarhus, Denmark, **4** Center for Eudaimonia and Human Flourishing, Linacre College, University of Oxford, Oxford United Kingdom, **5** Department of Psychiatry, University of Oxford, Oxford United Kingdom, **6** Institute of Computer Science, University of Bern, Bern, Switzerland, **7** Center for Experimental Neurology, Sleep Wake Epilepsy Center, NeuroTec, Department of Neurology, Inselspital, Bern University Hospital, University of Bern, Bern, Switzerland

* david.quiroga@psy.ku.dk

## Abstract

Vividly imagining a song or a melody is a skill that many people accomplish with relatively little effort. However, we are only beginning to understand how the brain represents, holds, and manipulates these musical "thoughts." Here, we decoded perceived and imagined melodies from magnetoencephalography (MEG) brain data (N = 71) to characterize their neural representation. We found that, during perception, auditory regions represent the sensory properties of individual sounds. In contrast, a widespread network including fronto-parietal cortex, hippocampus, basal nuclei, and sensorimotor regions hold the melody as an abstract unit during both perception and imagination. Furthermore, the mental manipulation of a melody systematically changes its neural representation, reflecting volitional control of auditory images. Our work sheds light on the nature and dynamics of auditory representations, informing future research on neural decoding of auditory imagination.

## 1. Introduction

Imagine your friends throwing a birthday party for you. At the climax, you begin to hear the first sounds of a well-known tune. "Happy birthday to you…", they cheerfully sing while you blow the candles and slice a delicious cake. If you are like most people, you can vividly recall the tune that your friends sing for you [1]. You may even recall the voice of a cherished friend or the crowd singing painfully out of tune. Yet, we are only beginning to understand how the brain represents, holds, and manipulates these musical thoughts [2].

Here, we consider 2 kinds of auditory imagination: Recall and manipulation. During **recall**, we accurately imagine previously known sounds. During **manipulation**, we imagine a modified version of the original sounds. In the brain, recall engages a widespread network including superior temporal gyrus, motor cortex, supplementary motor area, thalamus, parietal lobe, and frontal lobe [3–17], while manipulation further involves the frontal and parietal lobes

**Data availability statement:** Data are available in supporting files and the following online repository: https://doi.org/10.5281/zenodo.13760720. Materials and analysis code are available in this repository: https://doi.org/10.5281/zenodo.13760787.

**Funding:** This work was supported by NINDS R37NS21135 (RTK), Brain Initiative (U19NS107609-03 and U01NS108916) (RTK), CONTE Center PO MH109429 (RTK), the Independent Research fund, Denmark (DQM), the Carlsberg Foundation (CF23-1491) (DQM) and (CF20-0239) (LB), Lundbeck Foundation (Talent Prize 2022) (LB), Linacre College of the University of Oxford (Lucy Halsall fund) (LB), Nordic Mensa Fund (LB), the Danish National Research Foundation (DNRF 117) (GFR, LB, PV) , Mutua Madrileña Foundation (GFR), and the Interfaculty Research Cooperation "Decoding Sleep: From Neurons to Health & Mind" of the University of Bern and the Swiss National Science Foundation (#320030_188737) (AT). The funders had no role in study design, data collection and analysis, decision to publish, or preparation of the manuscript.

**Competing interests:** The authors have declared that no competing interests exist.

**Abbreviations:** ECG, electrocardiogram; EEG, electroencephalography; EOG, electrooculogram; fMRI, functional magnetic resonance imaging; GMSI, Goldsmiths Musical Sophistication Index; IRB, Institutional Review Board; LCMV, linearly constrained minimum variance; MEG, magnetoencephalography; MVPA, multivariate pattern analysis; ROI, region of interest; WAIS, Wechsler Adult Intelligence Scale.

[18,19]. With the exception of the visual cortex, these brain areas are largely consistent with those engaged in visual imagery [20–22]. However, it is unclear how these regions represent imagined sounds. By **representation**, we mean the neural activity patterns that distinguish an auditory object from others. Understanding neural representations is crucial for elucidating how the brain recreates and transforms auditory images in the mind's ear.

A powerful technique to study auditory representations is **multivariate pattern analysis** (MVPA) [23], where patterns of neural activity are used to decode features of mentally held objects. If neural signals carry object-specific information, decoding accuracy is different from chance. By inspecting decoding model coefficients, it is possible to identify the features of neural activity that underlie mental representations. Using similar techniques, functional magnetic resonance imaging (fMRI) studies showed sound-specific representations in primary and secondary auditory cortex [24–27] and frontoparietal association areas [28,29] during maintenance in working memory and imagination. Other studies demonstrated imagined sounds decoding from scalp electroencephalography (EEG) [30–32]. However, it remains unclear (**1**) how sound sequences are represented in auditory and association areas; (**2**) how these representations evolve in time; and (**3**) how they change when mentally manipulated.

Here, we used MVPA of brain activity recorded with magnetoencephalography (MEG, **Fig 1A**) to investigate how perceived, imagined, and mentally manipulated short auditory sequences are represented in the brain. For each trial in the task, participants listened and then were instructed to vividly imagine a short three-note melody (**Fig 1B**). In the recall block, participants imagined the melody as presented, whereas in the manipulation block they imagined it backwards (e.g., A-C#-E becomes E-C#-A). After a delay, they heard a second melody, which was the same as the first one, its backward version or a totally different one. Participants answered whether the second melody was the same as the first one or not (recall block) or the inverted version of the first one or not (manipulation block). Importantly, there were only 2 melodies to imagine in the task, which were backward versions of each other.

We first used a time generalized decoding technique [23] to characterize the neural dynamics of auditory representations. Then, we assessed whether mentally manipulating the melodies changed their neural representation. In the manipulation block, participants suppressed the forward pattern and mentally reinstated its backward version. Therefore, we predicted below-chance performance when training on manipulation and testing on recall and vice versa. Finally, we inspected model coefficients to identify the brain regions and neural activity features that discriminate between melodies and assessed how they changed between listening and imagination.

## 2. Results

### 2.1. Behavior

Participants ($N = 71$, 44 female, age = 28.77 +/− 8.43 SD) performed with high accuracy (**Fig 1C**) and were better ($OR = 1.85$, CI = [1.25–2.72], $p = 0.002$) in the recall (96.7%, CI = [95.6–97.6]) than the manipulation (94.1%, CI = [91.9–95.8]) block. Incorrect trials were excluded from MEG analyses. After the experiment, participants rated task-related imagery vividness on a 7-point Likert scale (from −3 to 3), with 72% of them rating 0 or above, which ranges from mild to strong vividness (Table 1). The good task performance and the vividness ratings suggest the presence of melody-specific information during imagination. Behavioral accuracy was associated with general working memory skills [33] (Wechsler Adult Intelligence Scale–WAIS; recall: r(69) = 0.3, $p = 0.012$; manipulation: r(69) = 0.29, $p = 0.016$; **Fig A** in **S1 Appendix**). No significant relationship with music training was found [34] (Goldsmiths Musical Sophistication Index–GMSI; recall: r(69) = 0.13, $p = 0.28$; manipulation: r(69) = 0.23, $p = 0.063$; **Fig B** in **S1 Appendix**; see also **Fig F** and **G** in **S1 Appendix** for further exploratory

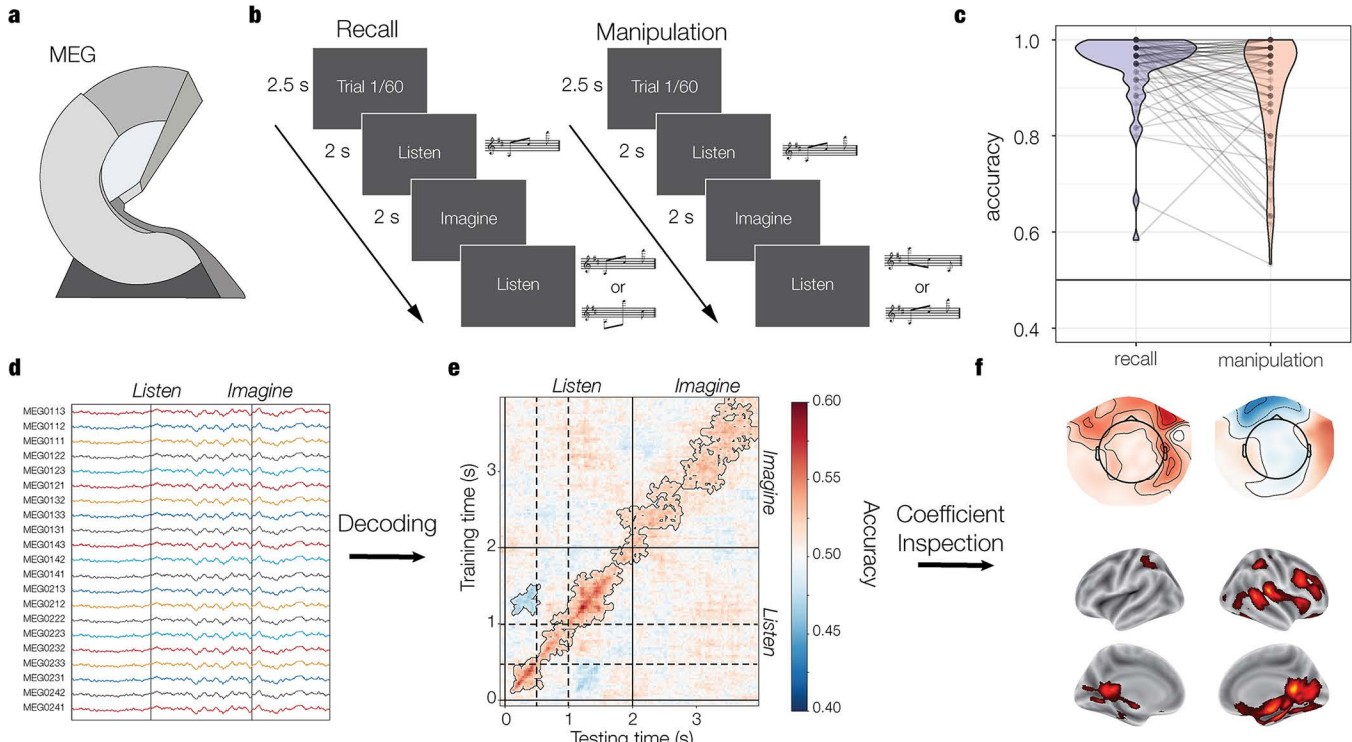

**Fig 1. Materials and methods.** We used MEG (a) to record the brain activity of 71 participants while they performed an imagery task (b). On each trial, participants heard and then imagined a short three-note melody. In the recall block, they imagined the melody as presented, while in the manipulation block, they imagined it backwards. Afterwards, they answered whether a test melody was the same as the first one (recall) or its backward version (manipulation). Participants performed with high accuracy (c) in both blocks. See S1 Fig for data related to this figure. MEG signals (d) were used to decode melody identity. We used a time-generalization approach (e) in which models were trained at each time point of the training trials and tested at each time point of the test trials, resulting in time-generalized accuracy matrices. We transformed model coefficients into patterns of activation (f) and localized their brain generators. Dashed lines mark the onset of the second (0.5 s) and third (1 s) sounds of the melodies.

analyses on possible associations of neural decoding with behavioral accuracy, vividness ratings, and music training).

## 2.2. Above-chance decoding of perceived and imagined melodies

To investigate the neural dynamics of musical representations, we trained logistic regression models on MEG sensor data (**Fig 1D**) to classify melody identity (melody 1: ***A-C#-E*** versus melody 2: ***E-C#-A***) at each time point of the trials. To assess whether representations recurred over time, we evaluated the models at each time point of the test data, resulting in time-generalized accuracy matrices (**Fig 1E**). We used 2 types of testing: **Within-condition** (training and testing on recall trials or training and testing on manipulation trials) and **between-condition** (training on manipulation trials and testing on recall trials or training on recall trials and testing on manipulation trials). The latter aimed to reveal whether mentally manipulating the melodies changed their neural representations.

**Table 1. Vividness ratings at the end of the task across participants.**

| Rating | −3 | −2 | −1 | 0 | 1 | 2 | 3 |
|---|---|---|---|---|---|---|---|
| Number of participants | 1 | 4 | 8 | 10 | 25 | 12 | 4 |
| Percentage | 1% | 6% | 11% | 14% | 35% | 17% | 6% |

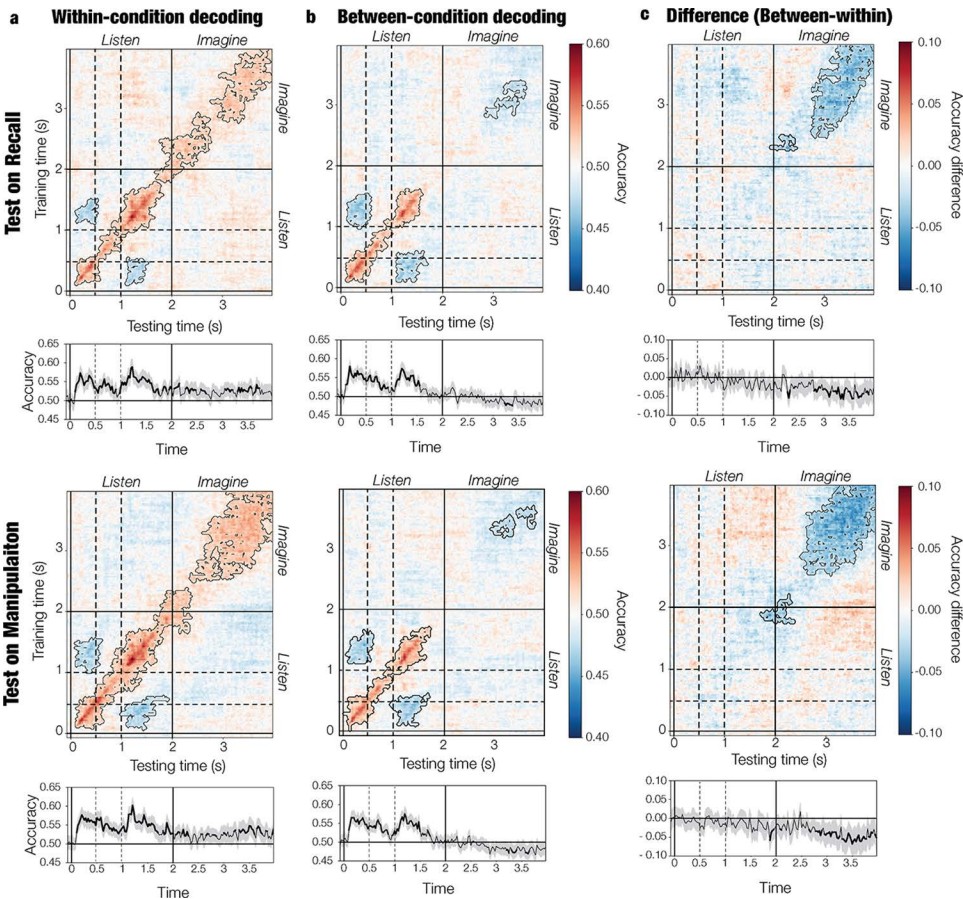

**Fig 2.** Accuracy for (a) within-condition decoding (train and test in the same condition), (b) between-condition decoding (train in one condition and test on the other), and the difference between the two (c). A time-generalization technique was used in which models were trained at each time point of the training data and tested at each time point of the test data. Accuracy across the diagonal is shown at the bottom of each plot. Contours and bold segments highlight significant clusters of above-chance or below-chance accuracy. Note how between-condition testing yields below-chance accuracy during imagination, suggesting a flip in neural representations. Dashed lines mark the onset of the second (0.5 s) and third (1 s) sounds of the melodies. See **S2 Fig** and the online repository for data and statistical outputs related to the current figure.

We observed above-chance within-condition decoding during listening and imagination for both recall and manipulation ($p < 0.001$; **Fig 2A**; see **Table B** in **S1 Appendix** for full statistical report). This further confirms that mental representations were present during the imagination period. Furthermore, we observed below chance performance when training around 0.3 s and testing around 1.3 s and vice versa reflecting the fact that the first sound (starting at 0 s) in one melody was the third sound (starting at 1 s) in the other melody ($p \leq 0.016$). This indicates that sound-specific representations discriminated between melodies at these time points. Note that the second sound (C#) was always the same.

## 2.3. Volitional control over imagined melodies

We used between-condition testing to decode the identity of the perceived melody at all time points in the trial and detect manipulation-related changes in neural representations. Thus, if during manipulation participants inhibited the representation of the perceived melody and reinstated the representation of its backward version, between-condition tests should

systematically predict the opposite of the perceived melody, resulting in below-chance accuracy in the imagination period. Indeed, we found below-chance accuracy both when training on recall and testing on manipulation ($p \leq 0.048$) and when training on manipulation and testing on recall ($p \leq 0.018$; **Fig 2B**; **Table A** in **S1 Appendix**). In both cases, accuracies were lower for between-condition than within-condition testing ($p \leq 0.035$; **Fig 2C**). This indicates a flip in neural representations such that models trained in one condition consistently predicted the opposite when tested on the other condition.

We also considered the possibility that representational dynamics were different between recall and manipulation. Indeed, when models were trained in the imagination period (approximately 3.5 s) and tested on the listening period (approximately 1.2 s) or vice versa, within-condition accuracy was lower ($p \leq 0.033$) for manipulation than recall (**Fig C** in **S1 Appendix**). This may reflect the fact that, for the manipulation block, the representation of the first melody was inhibited, thus leading to lower generalization across listening and imagination. Overall, these findings indicate that, in the manipulation block, participants inhibited the perceived melody and reinstated its backward version, resulting in a flip of neural representations. This provides evidence of **volitional control** over mental auditory representations.

## 2.4. Musical sound sequences are represented in auditory, association, sensorimotor, and subcortical areas

To elucidate the brain regions and neural features that distinguish between the melodies, we transformed the model coefficients into interpretable patterns of activation as described in [35], and localized their brain generators (**Fig 1F**). The resulting patterns can be interpreted as the differences in neural activity that discriminate between melodies and underlie successful decoding. We focused on average brain activity at 4 different periods: 3 during listening (0.2 s–0.5 s, 0.7 s–1 s, and 1.2 s–1.5 s) and 1 during imagination (2s–4s). For listening, we chose 200 to 500 ms after onset of each sound, starting at accuracy peaks (0.2 s and 1.2 s) and including sustained activity until sound offset. For the imagination period, we included the whole time interval due to the lower signal to noise ratio, the lack of prominent peaks, and the inherent temporal variability of mental images.

**2.4.1. Auditory representations during listening.** Patterns of neural activity distinguished between melodies in several brain areas. For the first sound (**Fig 3A**), we found clusters of regions in both conditions ($p \leq 0.006$, see **Table C** in **S1 Appendix** for full statistical reports) with peak activity patterns in **auditory areas** such as superior temporal gyrus and Heschl's gyrus, but also, in **somatosensory** (postcentral gyrus) and **association areas** (fusiform, hippocampus, retrosplenial, posterior cingulate, angular gyrus, inferior parietal cortex; see **Tables D** and **E** in **S1 Appendix** for a full report of anatomical regions). In addition, activity patterns in another cluster in both blocks ($p < 0.001$) peaked at anteromedial (orbitofrontal, anterior cingulate), posteromedial (mid-posterior cingulate), and lateral (inferior, middle, and superior frontal gyri) **prefrontal cortex**, as well as insula, **motor cortex** (precentral gyrus), and **subcortical structures** including the basal nuclei (putamen, caudate, accumbens, pallidum) and the thalamus. Interestingly, after the second sound information about melody identity was present in association, sensorimotor, and subcortical structures ($p < 0.005$), but not in auditory areas (**Fig 3B**). This reflects the fact that the second sound is the same in both melodies, inducing similar sensory representations in superior temporal cortex while maintaining distinct melody-wise representations across the brain.

The same areas outlined above represented the melodies after the third sound ($p \leq 0.021$; **Fig 3C**). Crucially, representations flipped sign in auditory areas and anterior medial temporal areas ($p \leq 0.003$; **Fig 3D**) such that, during sound 1, melody 2 elicited more positive

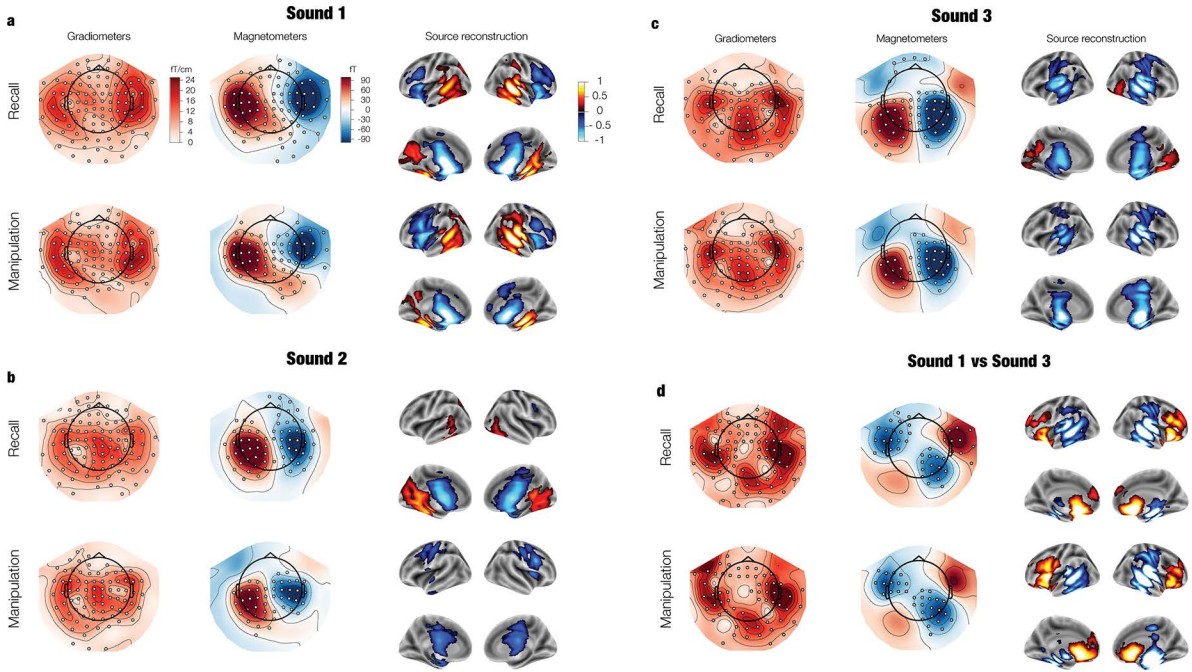

**Fig 3. Patterns of neural activity that discriminate between melodies during listening, as derived from decoding coefficients and averaged over time.** Patterns are shown for 3 time windows: (a) Sound 1 (0.2 s–0.5 s), (b) Sound 2 (0.7 s–1 s), and (c) Sound 3 (1.2 s–1.5 s). The difference between sounds 1 and 3 is also shown (d). Patterns are depicted for 2 types of MEG sensors (planar gradiometers and magnetometers) and after source reconstruction. For visualization, pairs of planar gradiometers were combined by taking their root mean square. Significant channels are highlighted with white dots. Source-level activation is shown for significant clusters. fT = "femto Tesla". See **S3 Fig** and the online repository for data and statistical outputs related to the current figure.

local field potentials than melody 1, whereas for sound 3 melody 1 elicited more positive local field potentials than melody 2 (**Fig D** in **S1 Appendix**). This representational flip underlies below-chance decoding after sounds 1 and 3 (**Fig 4A**) and reflects the fact that the 2 melodies are backward versions of each other. In addition, representations in the prefrontal cortex were more prominent after sound 1 ($p \leq 0.017$, **Fig 3D**) than sound 3, possibly indicating a more automatic evaluation at the end than at the beginning of the sequence [36,37]. Overall, these pieces of evidence suggest 2 types of processing: One concerned with **individual sound** encoding in auditory and anterior memory regions and another one concerned with holding the melody **as a sequence** in association, sensorimotor, and subcortical structures.

**2.4.2. Auditory representations during imagination.** In the imagination period, melodies were mainly represented in non-auditory areas including basal nuclei, thalamus, mid-posterior cingulate, motor, and parietal cortex ($p < 0.001$, **Fig 4A**). Additional recruitment of inferior temporal cortex, posterior cingulate, precuneus, and auditory areas was observed in the recall block, and of the lateral prefrontal cortex in the manipulation block. Furthermore, representations changed in the left lateral prefrontal cortex during manipulation compared to recall ($p = 0.033$; **Fig 4A**) with possible further changes in the right prefrontal cortex and retrosplenial (**Fig E** in **S1 Appendix**). These changes likely underlie the manipulation-driven representational flip identified through between-condition testing (**Fig 2B**).

## 2.5. Opposite neural activity during listening compared to imagination

Interestingly, patterns of activity switched sign ($p < 0.001$) between listening and imagination, with positive local fields in temporal areas becoming negative, and negative fields in anterior

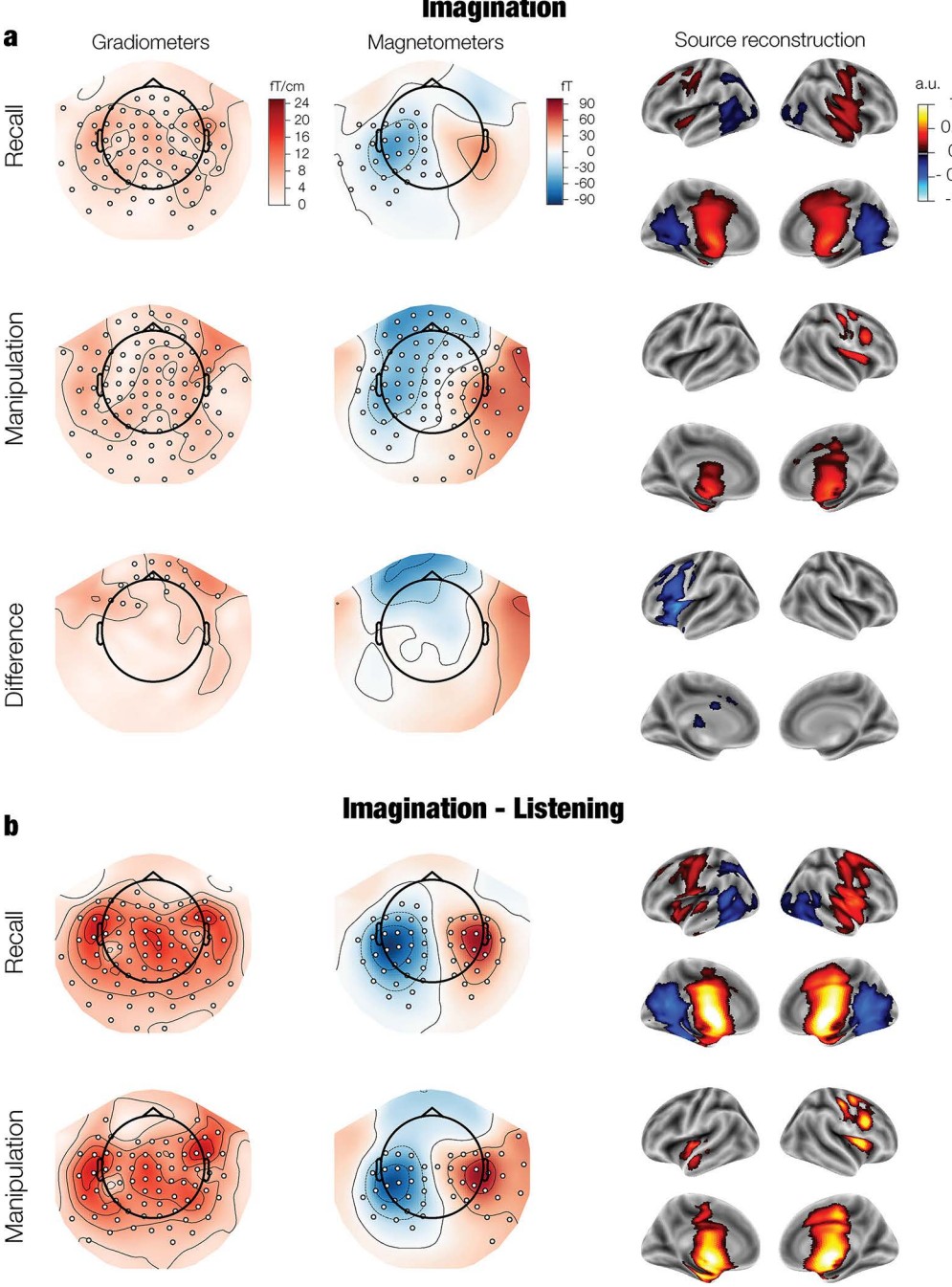

**Fig 4.** (a) Patterns of neural activity that discriminate between melodies during imagination in both conditions (2 s–4 s), as derived from decoding coefficients and averaged over time. The difference between listening and Imagination is also presented (b). Patterns are depicted for 2 types of MEG sensors (planar gradiometers and magnetometers) and after source reconstruction. For visualization, pairs of planar gradiometers were combined by taking their root mean square. Significant channels are highlighted with white dots. Source-level activation is shown for significant clusters. fT = "femto Tesla". See **S4 Fig** and the online repository for data and statistical outputs related to the current figure.

association, sensorimotor, and subcortical areas becoming positive after 2 s (**Figs 4B and D** in **S1 Appendix**). A similar switch was reported in studies that decoded imagined sounds from scalp EEG [30,31] and auditory working memory content with fMRI [25].

## 3. Discussion

In this study, we decoded perceived and imagined melodies from neural activity to demonstrate that musical sound sequences are represented in auditory, association, sensorimotor, and subcortical areas, and that these representations systematically change when mentally manipulated. While previous studies have decoded imagined sound information from brain data [24,26,29–32], here we define the nature and dynamics of the underlying auditory representations and show how they change during manipulation.

Above-chance decoding peaked after the onset of the first and third sounds and was sustained during imagination. The highest decoding performance was detected around the diagonal of the matrices, which indicates that representations were dynamic and had marginal generalization over time [23]. During listening, this could be due to the constantly changing sensory input. During imagination, this might reflect temporal variability between participants. The lack of generalization further suggests that representations were different between listening and imagination. This contrasts with research suggesting that perceived and imagined sounds share neural substrates and representations. For example, both imagined and actual sounds activate secondary auditory areas [13] and fMRI studies decoded imagined auditory representations from primary and secondary auditory cortex [24–27]. Moreover, some studies found that representations during the omission of predictable sounds are similar to those of the actual sounds [38,39].

The lack of generalization in our results might arise from 3 factors. First, the melodies differed in the temporal order of their constituent sounds, which were otherwise the same. Temporal order is an abstract feature that might generalize less across listening and imagination than sensory features such as pitch, which are typically the target of decoding (e.g., [40]). Second, experimental paradigms have either minimized the temporal variability of the imagined representations (e.g., omission studies) [38,39] or did not take time into account (fMRI studies) [24–27]. In contrast, our design allows temporal flexibility within a relatively long imagination period (2 s), which might introduce between-trial and between-subject variations that blur sound-specific representations. Finally, consistent with previous EEG and fMRI findings [25,30,31], representations in association, sensorimotor, and subcortical areas were opposite between listening and imagination (**Fig 3B**). This flip could reflect a change in the direction of information flow across the brain, from bottom-up (listening) to top-down (imagination). Some models propose specific roles for different layers of the cortical sheet, with superficial pyramidal cells conveying bottom up sensory input, and deep pyramidal cells conveying top-down expectations [41,42]. It is possible that these layer-specific efferent and afferent activity patterns result in detectable changes in the direction of local field potentials. Layer-sensitive recordings are needed to test this hypothesis.

The fact that the melodies were backward versions of each other allowed us to dissociate the role of different brain areas. During listening, auditory regions encoded the sensory information of individual sounds such that representations were opposite after the first and third tones, and equal during the second sound. In contrast, association and subcortical areas (inferior and medial temporal lobe, ventromedial prefrontal cortex, thalamus, and basal nuclei) remained stable, while representations in dorsal association areas (lateral prefrontal cortex) were involved only at sequence onset. Moreover, during imagination, association and subcortical areas were the main carriers of representations, with auditory and temporal areas further involved during recall, and lateral prefrontal cortex further engaged in manipulation. Overall, these dynamics suggest 2 types of processing: one concerned with the encoding of sound-specific sensory information in superior temporal cortex and anterior temporal areas, and another one concerned with the encoding, retrieval, and manipulation of auditory

sequences in association and subcortical areas. This dissociation between the sensory and abstract properties of sound sequences is consistent with a previous scalp EEG study that disentangled pitch and temporal order representations during sound maintenance in auditory working memory [43].

The regions that carried auditory representations in our study overlap with those identified in previous neuroimaging activation studies as important for imagery in audition and other modalities [4,6–22]. However, a discrepancy of our study is the lack of substantial melody-specific information in the supplementary motor area, identified as a key region for auditory imagination [44]. Nevertheless, we found representations in the motor and somatosensory cortex, which is consistent with previous reports [7,45,46] and might reflect the generation of auditory expectations through motor simulation. Furthermore, we observed melody-specific representations in the basal nuclei, a set of areas involved in both cognitive and motor control that have not been identified in previous auditory imagery research. From these nuclei, the putamen has been related to motor imagery [47]. Moreover, the basal nuclei are typically studied with the hemodynamic response in fMRI, which correlates best with high gamma (>60 Hz) power [48] in EEG. In this study, we used instead MEG broadband signals to decode auditory objects, which might be why basal nuclei representations were found here but not in fMRI. Future research examining high gamma activity and other frequency bands will be needed to elucidate their relationship with the hemodynamic response.

The task used in this study is similar to the classical delayed match-to-sample paradigm employed in working memory research. An important difference, however, is that we asked participants to vividly imagine and mentally manipulate the melodies, whereas in working memory experiments maintenance strategies usually remain unspecified. Thus, while it is possible that our participants used unconscious maintenance strategies without imagery, the explicit task instruction, the good task performance, the vividness ratings, and the between-condition decoding results suggest that they engaged in active mental recall and manipulation. Future experiments where imagery is not required are needed to further elucidate the nature of maintenance strategies and the relationship of imagery with working memory.

This caveat aside, task performance was associated with general working memory scores and the brain regions identified overlap with those exhibiting delay-period activity in auditory working memory, including the auditory cortex, the prefrontal cortex, the parietal cortex, and the medial temporal lobe [19,28,49,50]. Moreover, auditory representations in working memory have been decoded from the auditory, frontal, and parietal cortices [24,28,43] and from the functional interaction of these regions [51,52]. Most of these decoding studies, however, addressed working memory for individual sounds and none investigated sound manipulation. In addition, there is a tradeoff, with fMRI studies having good spatial but low temporal resolution, and EEG studies having good temporal but low spatial resolution. The use of MEG allowed a good localization of auditory representations both in space and time.

Two methodological caveats need to be considered. First, we localized auditory representations to both cortical and deep brain areas (basal nuclei, thalamus, and hippocampus), raising concerns given the bias towards the head center of beamforming algorithms [53] and the fact that activity in such areas is typically hard to detect with MEG. However, we eliminated the depth bias by normalizing the forward and inverse solutions and verified that the localized activations are consistent with sensor topographies, especially at the midline (e.g., **Fig 3C**). In addition, differences were still found in deep structures when 2 conditions were contrasted (e.g., imagination versus listening), arguing against a depth bias which should cancel out in condition contrasts. Furthermore, with implementation of appropriate controls, the use of beamformers has made the detection of deep sources increasingly common, including the basal

nuclei, the medial temporal lobe, and even the cerebellum [54–57]. Therefore, it is unlikely that these deep activity patterns are localization artifacts. The other caveat is the possibility that successful decoding is partly due to extracerebral, motion-related activity. However, this is also unlikely because we thoroughly cleaned the data from the main sources of contamination (eye movements and heartbeats), the sensor topographies suggest brain generators, and beamforming algorithms are particularly good at filtering out extracerebral sources.

In conclusion, our results provide evidence regarding the nature and dynamics of perceived and imagined sound representations in the brain and contribute to a growing body of work investigating musical imagery and its relationship with other modalities [2,58–63]. Our findings also demonstrate the feasibility of decoding mental auditory representations at a fine temporal resolution with noninvasive methods. This opens the path to clinical applications where decoding of imagined objects is relevant (e.g., communication impairments). Future work might employ different recording modalities (e.g., optical MEG, intracranial EEG), bigger data sets (e.g., by increasing the number of trials), and models that are larger and account for the temporal variability in imagination (e.g., deep learning) [64] to maximize the decoding of auditory images.

## 4. Methods

### 4.1. Participants

We recorded MEG (**Fig 1A**) data from 80 participants. From these, 6 were excluded due to chance behavioral performance and 3 due to noisy or corrupted neural data, resulting in a final group of 71 participants (44 female, age = 28.77 +/− 8.43 SD). Three of these participants were excluded from source level analyses due to absence of anatomical images. Participants had mixed musical backgrounds with most of them [50] never having played a musical instrument (including voice). The other 21 participants had a median of 11 (IQR = [7–16]) years of musical training. In addition, participants had a median score of 17 (IQR = [13–26], maximum possible score = 49) in the training subscale of the Goldsmiths Musical Sophistication Index (GMSI) [34] and of 96 (IQR = [93–105]) in the Wechsler Adult Intelligence Scale (WAIS) [33]. Musical expertise was not a factor in recruitment for this experiment. Participants gave written informed consent and received a small monetary compensation. The study was approved by the Institutional Review Board (IRB) of Aarhus University (case number: DNC-IRB-2020-006) and conducted in accordance with the Helsinki declaration.

### 4.2. Stimuli

We employed short three-note melodies forming a major chord arpeggio using piano sounds (musical pitch: A3, C#5, E6; F0: 220Hz, 554Hz, 1318Hz) synthesized with MuseScore (v3.6.2; see materials' online repository for the actual sounds used). The sounds were arranged in ascending order in melody 1 (A-C#-E) and descending order in melody 2 (E-C#-A). Two foil test melodies were also included: A-E-C# and E-A-C#. The inter-onset interval between individual sounds was 500 ms. The sounds were normalized to peak amplitude.

### 4.3. Task

The experiment was implemented in Psychopy v3.1.2 [65] (see materials' online repository for details). On each trial (**Fig 1B**), participants heard melody 1 or melody 2, together with the word "Listen" appearing on the screen. After 2 s, participants saw the word "Imagine," which indicated that they had to vividly reproduce the melody in their minds. There were 2 conditions, encompassing the 2 different blocks in the experiment. In the recall block, they imagined the melody as presented, whereas in the manipulation block, they imagined it

backwards. Four seconds after trial onset, participants heard a test melody, which could be the same as the first one, its inverted version or a different melody. Participants answered whether the second melody was the same as the first one or not (recall block) or its inverted version or not (manipulation block). A response time limit of 3.5 s was set. There were 60 trials per block (30 same/inverted, 30 different/other). The trial number was displayed on the screen for 2.5 s before trial onset. A quick pause was allowed after the 30th trial. Two practice trials were presented at the beginning of each block. Conditions were counterbalanced across participants.

### 4.4. Procedure

At the beginning of the session, we explained in detail the procedure to the participants and instructed them to vividly imagine the melodies without humming them or moving any part of the body. We made sure the participant fully understood the nature of the task and was able to perform practice trials correctly before the MEG recording. After giving written informed consent, the participants changed into medical clothes, and we attached electrocardiogram (ECG) and electrooculogram (EOG) electrodes to their skin for heartbeat and eye movement monitoring. Head shape was digitized with a Polhemus system and head position was continuously tracked during the recording with the help of 3 coils. During the task, the participant sat in the MEG chair inside a magnetically shielded room and looked at the screen where instructions and trial information were displayed. The subjects responded to each trial by making a button press in a response pad with their right hand. Sound stimulation was delivered through magnetically isolated ear tubes. The task lasted approximately 20 min. Other experimental paradigms testing recognition memory were recorded together with this task. Results are reported elsewhere [57]. The order of the paradigms was counterbalanced across participants. After the experiment, participants were asked to rate the vividness of imagery during the task on a 7-point Likert scale ranging from −3 to 3.

### 4.5. MEG recording and preprocessing

MEG data were collected with a 306-channel (102 magnetometers, 204 planar gradiometers) Elekta Neuromag system and Maxwell-filtered with proprietary software. This step also involved correcting the data for continuous head movements. Data analyses were conducted in MNE Python (v0.24) [66]. Vertical and horizontal eye movements as well as heartbeat artifacts were corrected with ICA in a semi-automatic routine. Visual inspection was used to ensure data quality. After high-pass filtering (0.05 Hz cutoff), epochs were extracted from −0.1 s to 4 s around trial onset. For source reconstruction, T1 brain anatomical images were collected with a 3T MRI scanner and segmented and aligned with MEG sensors using Freesurfer. Source reconstruction was done for the 68 participants with an available MRI. Using the boundary element method and a single shell mesh (5 mm resolution), volumetric forward models were created and subsequently inverted with linearly constrained minimum variance (LCMV) beamforming employing the joint gradiometer covariance across listening and imagination periods. For similar results obtained with the separate covariance of the listening and imagination periods, see **Fig H** in **S1 Appendix.** Importantly, forward models and inverse solutions were normalized to eliminate the bias towards the center of the head inherent to beamformers [53].

### 4.6. Decoding analysis

We used a time-generalized decoding approach (Fig 1D) [23] based on L1 regularized logistic regression to classify melody identity (melody 1 versus melody 2) at each time point of the trials, for each participant separately. To assess the representational dynamics, we evaluated the

models at each time point of the test data. We did 2 types of testing. In within-condition testing, we trained and tested the models with trials of the same condition. In between-condition testing, we trained the models with trials of one condition (e.g., manipulation) and tested on trials of the other (e.g., recall). Five-fold cross validation was used for within-condition testing. To avoid biases in model fitting due to class imbalances related to the exclusion of incorrect trials, we used a balanced scoring strategy in which the average accuracy was computed separately for each class and then combined across classes.

At the group level, we used nonparametric cluster-based permutations [67] to evaluate whether accuracies in the time-generalization matrices were significantly above or below chance. Here, chance level corresponds to 0.5 accuracy, as we classified binary melody identity from brain data. We used a two-sided cluster-defining threshold of $p = 0.05$ based on one-sample $t$ tests ($p = 0.025$ one-sided, $t > 1.99$) and max sum as the cluster statistic. The cluster-level significance threshold was set at $p = 0.05$. The number of permutations was 5,000. The same statistical approach was used to evaluate whether within-condition accuracy was different from between-condition accuracy.

### 4.7. Coefficient inspection

We transformed decoding coefficients ($\mathbf{W}$) into interpretable patterns of activation ($\mathbf{A}$) for each participant using the method detailed in [35] and defined by the equation:

$$\mathbf{A} = \Sigma_x \mathbf{W} \Sigma_{\hat{y}}^{-1}.$$

Where $\Sigma_{\hat{y}}$ is the covariance of model predictions and $\Sigma_x$ is the covariance of neural signals.
We localized the neural generators of these patterns using the inverse solutions described in section 4.5 (**Fig 1F**). For each voxel, the magnitude and sign of the orientation with maximum power were retained. For sensor activity patterns, we used cluster based permutations (see above) in the whole epoch (0–4) to test whether group-level activity patterns were different from zero. After projecting individual source-level time courses into MNI standard space, we also tested against zero the localized patterns averaged across time in the 3 listening (0.2s – 0.5s, 0.7s – 1s, 1.2s – 1.5s) and 1 imagination (2s – 4s) time windows. For all these periods, differences between recall and manipulation were also tested. Furthermore, we compared patterns of average activity between the listening (0s – 2s) and imagination (2s – 4s) periods and between sounds 1 (0.2s – 0.5s) and 3 (1.2s – 1.5s). Using the Desikan–Killiany parcellation [68], we obtained the significant peak activation for each region that overlapped with significant clusters. We report the regions with the most prominent peaks.

Finally, in a supplemental analysis, we inspected the time courses of activity patterns in 5 groups of regions of interest (ROIs) including 1) right auditory, 2) right posteroventral association, 3) right dorsal association, 4) left dorsal association, and 5) right anteroventral/subcortical areas (**Fig D** in **S1 Appendix**). We used cluster based permutations as described above to evaluate significant differences from zero. We display these patterns together with the evoked response calculated between –0.1 s and 4 s around trial onset, for each of the 2 melodies and the 2 conditions (**Fig D** in **S1 Appendix**). These evoked responses were source localized with the same inverse operator as the activity patterns derived from decoding and were subject to the same statistical tests.

## Supporting information

**S1 Appendix. Supplemental Figures and Tables.**
(PDF)

**S1 Fig. Data for Fig 1.**
(XLSX)

**S2 Fig. Data for Fig 2.**
(XLSX)

**S3 Fig. Data for Fig 3.**
(XLSX)

**S4 Fig. Data for Fig 4.**
(XLSX)

**S5 Fig. Data for Fig F in S1 Appendix.**
(XLSX)

**S6 Fig. Data for Fig G in S1 Appendix.**
(XLSX)

## Acknowledgments

We thank Francesco Carlomagno for his help with data collection and Ludovic Bellier for feedback.

## Author contributions

**Conceptualization:** David R. Quiroga-Martinez, Leonardo Bonetti, Athina Tzovara, Robert T. Knight.

**Data curation:** David R. Quiroga-Martinez, Gemma Fernández Rubio, Leonardo Bonetti, Kriti G. Achyutuni.

**Formal analysis:** David R. Quiroga-Martinez, Gemma Fernández Rubio.

**Funding acquisition:** David R. Quiroga-Martinez.

**Investigation:** David R. Quiroga-Martinez.

**Methodology:** David R. Quiroga-Martinez, Leonardo Bonetti.

**Project administration:** David R. Quiroga-Martinez.

**Resources:** Robert T. Knight.

**Software:** David R. Quiroga-Martinez.

**Supervision:** Athina Tzovara, Robert T. Knight, Peter Vuust.

**Validation:** David R. Quiroga-Martinez.

**Visualization:** David R. Quiroga-Martinez, Gemma Fernández Rubio, Kriti G. Achyutuni.

**Writing – original draft:** David R. Quiroga-Martinez.

**Writing – review & editing:** David R. Quiroga-Martinez, Athina Tzovara, Robert T. Knight, Peter Vuust.

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
