## [Editor Report · Decision Letter 0]

9 May 2024

Dear Dr Quiroga-Martinez, 

Thank you for submitting your manuscript entitled "Decoding reveals the neural representation of perceived and imagined musical thoughts" for consideration as a Research Article by PLOS Biology.

Your manuscript has now been evaluated by the PLOS Biology editorial staff as well as by an academic editor with relevant expertise and I am writing to let you know that we would like to send your submission out for external peer review. Please note that, while we discussed your manuscript with one of our academic editors, we are not completely sure whether the conceptual advance is sufficient for PLOS Biology. We will, therefore, be looking for strong enthusiasm from the reviewers.

Once your full submission is complete, your paper will undergo a series of checks in preparation for peer review. After your manuscript has passed the checks it will be sent out for review. To provide the metadata for your submission, please Login to Editorial Manager (https://www.editorialmanager.com/pbiology ) within two working days, i.e. by May 11 2024 11:59PM.

Kind regards,

Christian

Christian Schnell, PhD

Senior Editor

PLOS Biology

cschnell@plos.org

---

## [Decision Letter · Decision Letter 1]

1 Aug 2024

Dear Dr Quiroga-Martinez,

Thank you for your patience while your manuscript "Decoding reveals the neural representation of perceived and imagined musical thoughts" was peer-reviewed at PLOS Biology. It has now been evaluated by the PLOS Biology editors, an Academic Editor with relevant expertise, and by several independent reviewers. 

In light of the reviews, which you will find at the end of this email, we would like to invite you to revise the work to thoroughly address the reviewers' reports.

As you will see below, the reviewers think that the study is well executed and provides important insights. Reviewer 1 has some concerns about the experimental design and a missing control group missing. Reviewer 2's requests are mostly about a better integration of the current manuscript into the existing literature and textual clarifications. Reviewer 3 has submitted some very minor comments. 

Given the extent of revision needed, we cannot make a decision about publication until we have seen the revised manuscript and your response to the reviewers' comments. Your revised manuscript is likely to be sent for further evaluation by all or a subset of the reviewers.

We expect to receive your revised manuscript within 3 months. Please email us (plosbiology@plos.org ) if you have any questions or concerns, or would like to request an extension. 

**IMPORTANT - SUBMITTING YOUR REVISION**

*Re-submission Checklist*

*Published Peer Review*

*PLOS Data Policy*

Please note that as a condition of publication PLOS' data policy (http://journals.plos.org/plosbiology/s/data-availability ) requires that you make available all data used to draw the conclusions arrived at in your manuscript. If you have not already done so, you must include any data used in your manuscript either in appropriate repositories, within the body of the manuscript, or as supporting information (N.B. this includes any numerical values that were used to generate graphs, histograms etc.). For an example see here: http://www.plosbiology.org/article/info%3Adoi%2F10.1371%2Fjournal.pbio.1001908#s5

*Blot and Gel Data Policy*

Sincerely,

Christian

Christian Schnell, PhD

Senior Editor

PLOS Biology

cschnell@plos.org

REVIEWS:

Reviewer #1: This study aims to characterize the neural representations of imagined melodies. Participants are tasked to recall and manipulate simple three-note melodies while their MEG is measured. Using a combination of multivariate pattern analysis and source reconstruction, the authors find significant representations of imagined (both recalled and manipulated) melodies throughout several association areas. Furthermore, interesting sign flips are found between perception and imagination, providing important avenues for future research. 

The paper is well-written and uses advanced analysis in a comprehensive way to study an important question about the neural mechanisms of auditory imagination. However, I do have several methodological queries that need to be addressed. 

Major comments:

Due to the experimental design, in the recall block, imagine A is always preceded by listen A and imagine B is always preceded by listen B. Therefore, is it possible that (part of) the results are due to temporal autocorrelations in the signal between listening and imagining? I appreciate that this cannot explain the between-condition negative decoding, but could it explain within-condition decoding as well as part of the source analysis results? 

The behavioural results of a ceiling effect are taken to 'confirm the presence of melody specific information during the imagination period'. However, unless I missed something, it could just as well reflect accurate memory of the 'listen' period, right? Especially as the accuracy is higher during the recall compared to the manipulation blocks. Relatedly, without a control condition in which participants were not explicitly asked to 'imagine as vividly as possible' it is hard to know which of the results reflect conscious imagery versus (potentially unconscious) working memory. This is important because recent studies with people who are unable to imagine (aphantasia) suggest that the two might rely on different neural mechanisms (see e.g. Keogh et al., 2021 Cortex).

It was unclear to me how exactly the decoding weights/coefficients were transformed to 'activity' patterns for source localisation. In most cases, decoding weights do not only reflect which sensors/sources contribute stimulus information, but large weights can also be obtained even if the associated sensors/sources do not contain stimulus info, e.g. because they are used for noise suppression by the decoding algorithm. More explanation is necessary here to ensure that the authors have transformed the decoding coefficients accurately. For an extensive discussion on this topic, see Haufe et al., 2014 NeuroImage. 

What does 'permutation-corrected' mean in reference to testing for significance of decoding? In general, decoding accuracy ois known not to meet all the assumptions of parametric statistical testing and therefore permutation-based methods are more suitable, for more info, see Stelzer et al., 2013 NeuroImage. 

Minor comments:

Please mention in the legend that the dashed lines in Fig. 1F reflect the onset of the different tones. 

I applaud the large sample size but it also made me wonder whether this was initially set-up as a group-study, given that a portion of the participants had zero musical training whereas others had a median of 11 years musical training. Is that the case? And if so, please report this. Out of interest, were there any differences in decoding depending on musical training? 

How many blocks were there in total?

It took me a while to understand that the manipulation block trials were labelled based on the listening period, not on the imagination period, and that below-chance accuracies were therefore expected. It would help the reader if this was made more explicit and clearer in the text. 

Please mention under the participants heading that three additional participants were excluded for source analysis due to unavailable MRI's.

Why did you use logistic regression for decoding rather than the more common linear regression? 

Reviewer #2: Dear Dr. Schnell,

Dear Authors,

I thoroughly enjoyed reviewing "Decoding reveals the neural representation of perceived and imagined musical thoughts" submitted for publication in "PLOS Biology" (review request received: 27 June 2024, review accepted: 01 July 2024, review completed: 08 July 2024).

This research opens a new door in understanding the perceived and imagined melodies from MEG data and is novel in its contribution to the current field researching imagination, music, and the brain. Particularly, MVPA analysis adds a new flavour to this investigation, which contributes to other neuroimaging techniques being employed, specifically studying the auditory cortex.

In general, the work is very well written, is effective in conveying the information directly to the reader and fits within the scope of "PLOS Biology", and I recommend this manuscript for publication as it presents a valuable impact to the field. It is already in great shape; however, I have some comments that I recommend the authors should address in revision. I intend these comments as constructive feedback to support and enhance your efforts.

Comments

Abstract

Comment: "With little effort you could sing it or play it vividly in your mind." I find this sentence too generalised. While many people can do this, this ability may not come naturally to everyone. I guess with could, you tried to convey this, however, I believe rephrasing it may enhance clarity.

Introduction

Comment: The "Happy birthday" example is very commonly used. It is effective, but reference(s) are needed here, e.g.,

Cotter, K. N., Christensen, A. P., & Silvia, P. J. (2019). Understanding inner music: A dimensional approach to musical imagery. Psychology of Aesthetics, Creativity, and the Arts, 13(4), 489.

Küssner, M. B., Taruffi, L., & Floridou, G. A. (Eds.). (2023). Music and Mental Imagery. London and New York: Routledge.

Halpern, A. R. (2012). Dynamic aspects of musical imagery. Annals of the New York Academy of Sciences, 1252(1), 200-205. 

Poliakoff, E., Bek, J., Phillips, M., Young, W. R., & Rose, D. C. (2023). Vividness and Use of Imagery Related to Music and Movement in People with Parkinson's: A Mixed-methods Survey Study. Music & Science, 6, 20592043231197919.

Di Liberto, G. M., Marion, G., & Shamma, S. A. (2021). Accurate decoding of imagined and heard melodies. Frontiers in Neuroscience, 15, 673401.

Comment: p. 13, fMRI and EEG to be spelled out the first time.

Comment: Introduction lacks literature that is current and related to this study, such as auditory imagery with or without MEG. For instance, very relevant works include:

Herff, S. A., Herff, C., Milne, A. J., Johnson, G. D., Shih, J. J., & Krusienski, D. J. (2020). Prefrontal High Gamma in ECoG tags periodicity of musical rhythms in perception and imagination. Eneuro, 7(4).

Dash, D., Ferrari, P., & Wang, J. (2020). Decoding imagined and spoken phrases from non-invasive neural (MEG) signals. Frontiers in neuroscience, 14, 290.

Cheng, T. H. Z., Creel, S. C., & Iversen, J. R. (2022). How do you feel the rhythm: Dynamic motor-auditory interactions are involved in the imagination of hierarchical timing. Journal of Neuroscience, 42(3), 500-512.

Schaefer, R. S. (2014). Images of time: Temporal aspects of auditory and movement imagination. Frontiers in Psychology, 5, 877.

Dijkstra, N., Kok, P., & Fleming, S. M. (2022). Perceptual reality monitoring: Neural mechanisms dissociating imagination from reality. Neuroscience & Biobehavioral Reviews, 135, 104557.

Dijkstra, N., & Fleming, S. M. (2023). Subjective signal strength distinguishes reality from imagination. Nature Communications, 14(1), 1627.

Intro can also benefit from more primary music-related imagery work that is associated with your work, as there are many imagery modalities - auditory, vision -, and this body of literature below, for instance, is currently investigating specific/overall modalities. It is worth pointing out here what imagery modalities you exactly are investigating. If that is not modality-specific, that is also okay, but I cannot be exactly sure whether it is auditory or other modalities as well that you are investigating here from the text, or the definition of the "imagination" is not too clear. 

Küssner, M. B., & Taruffi, L. (2022). Modalities and causal routes in music-induced mental imagery. Trends in Cognitive Sciences.

Taruffi, L., Ayyildiz, C., & Herff, S. A. (2023). Thematic Contents of Mental Imagery are Shaped by Concurrent Task-Irrelevant Music. Imagination, Cognition and Personality, 43(2), 169-192.

Herff, S. A., Cecchetti, G., Taruffi, L., & Déguernel, K. (2021). Music influences vividness and content of imagined journeys in a directed visual imagery task. Scientific Reports, 11(1), 15990.

Herff, S. A., McConnell, S., Ji, J. L., & Prince, J. B. (2022). Eye Closure Interacts with Music to Influence Vividness and Content of Directed Imagery. Music & Science, 5, 20592043221142711.

Margulis, E. H., & Jakubowski, K. (2024). Music, Memory, and Imagination. Current Directions in Psychological Science, 33(2), 108-113.

Margulis, E. H., & McAuley, J. D. (2022). Mechanisms and individual differences in music-evoked imaginings. Trends in Cognitive Sciences.

Comment: "However, no study has elucidated 1) the nature of sound sequence representations in auditory and association areas, 2) how these representations evolve over time and 3) how they change when mentally manipulated." - "no study" is simply untrue and very argumentative within this sentence as there are many studies looking at these aims (such as above and some of the work you've already cited). Although your study is novel in its aims noted elsewhere, please change the wording here or change the sentence in a way that no study studied what you exactly are looking at here.

Results

Comment: "Participants (N = 71, 44 female, age = 28.77 +/- 8.43 SD) performed with high accuracy (Fig. 1c) and were better (OR = 1.85, CI = [1.25 - 2.72], p = .002) in the recall (96.7%, CI = [95.6 - 97.6]) than the manipulation (94.1%, CI = [91.9 - 95.8]) block. This confirms the presence of melody specific information during the imagination period." - This finding can also be interpreted in other ways, such as differences in task difficulty or participant engagement between the recall and manipulation blocks that might be crucial to understand the differences of the recall and manipulation blocks. Feel free to ignore this, but maybe, it is worth pointing out in the discussion.

Comment: "This is indicative of a flip in neural representations such that models trained in one condition were consistently "wrong" when tested on the other condition." - Here, I feel "wrong" is the wrong word to use here. Possibly, "inaccurate" might be a better word as it is more precise and scientifically appropriate in the context of neural representations and model testing. 

Comment: Sound and tones are interchangeably used. It is worth keeping these terms consistent. 

Discussion

Comment: 2nd and 3rd paragraph of the discussion could be combined as the 2nd paragraph feels incomplete, and they seem to be very short paragraphs. There are missing references too, and it is unclear whether the last sentences are speculation or not to the general reader, so references would go a long way here.

Comment: "Temporal order is an abstract feature that might generalize less across listening and imagination than sensory features such as pitch, which are typically the target of decoding."- missing reference here. E.g.,

Sankaran, N., Thompson, W. F., Carlile, S., & Carlson, T. A. (2018). Decoding the dynamic representation of musical pitch from human brain activity. Scientific reports, 8(1), 839.

Comment: I'd also include references to the omission studies examples as well as for the fMRI ones.

Comment: "…which might introduce between-trial and between-subject variations that blur sound-specific representations.". - I think it could be important to highlight how this could be addressed, such as including more trials to enhance the reliability of the data, or considering using mixed-effects models to account for both fixed (conditions) and random (participant variability). I guess running mixed-effects models could address this limitation most straightforwardly within this paper, but that is up to the authors' choice, as simply addressing the limitation and how to address it in the future is also okay.

Comment: "previous neuroimaging activation studies as important for auditory imagery." - Is this only relevant for auditory imagery or other modalities too? Or is there something special related to auditory imagery here? 

Comment: "Furthermore, we observed melody specific representations in the basal nuclei, a set of areas involved in both cognitive and motor control that have not been identified in previous fMRI research" - Here, I find "previous fMRI research" very specific. It might be worth clarifying which fMRI research, such as including "… fMRI research on melody processing." And I might be missing something here, but why only fMRI research? How about EEG, etc., or is this only an example?

Comment: "Finally, our results demonstrate the feasibility…". - I would change "Finally" to "Overall" as finally read like it is the final point you want to make. This paragraph could take advantage of adding a sentence regarding the purpose of the study and how this demonstrated in your results.

Method

Comment: For stimuli, no need for "timbre" for piano, as this can be confusing to the general reader if the definition of timbre is not being used before (as it is a complex subject and definition is ill-defined and in debate (Siedenburg et al., 2019), just leaving "piano" would be sufficient.

Comment: The stimuli section is too brief. It lacks information on whether the stimuli are recorded (if it was, how?), or taken from elsewhere, or produced/ artificially made? Did the piano sound natural or not (e.g., artificially made)? Currently, one cannot replicate re-producing the stimuli. 

Comment: melody1 and melody2 are written as melody 1 and melody 2 at different points of the Method section. 

Comment: Regarding the task, how long did the experiment take? 

Comment: "using the inverse solutions described above (Fig. 1f)"- in Figure 1f, instead of above.

Comment: For completeness, I liked the addition of relationship with behaviour. Is there a missing citation here: "(R, lme4; , 52, 53)? However, it might also be worth adding this section to the supplementary materials instead of here as this does not significantly add to the main points. Words could be used more to improve the Introduction and Discussion section.

Comma Comments:

- Oxford comma inconsistency

- Missing commas (e.g., "Interestingly, after the second sound, melodies were discriminated in association, sensorimotor and subcortical structures (p < .005), but not in auditory areas (Fig. 3b).

Thank you a lot for your fascinating contribution, and I look forward to seeing a revised and published version of this study.

Reviewer #3: This is a well conducted study that provides a valuable contribution to the literature. I have only the following brief comments 

It is not clear what is meant with "This confirms the presence of melody specific information during the imagination period". How is performing accurately conclusive evidence of imagination during the imagination period?

As per above, I would suggest that "This further confirms that mental representations were present during the delay period." be amended to "This confirms that mental representations were present during the delay period" Also note that neither "imagination period" or "delay period" are introduced clearly. Presumably they are the same? Introduce one and stick to it.

In "We also considered the possibility that representational dynamics were different between recall and manipulation. This is interesting because, during manipulation, the representation of the listened melody is meant to be the opposite of the imagined melody, potentially yielding different temporal generalization patterns.", make clear WHAT is interesting. The possibility is interesting?

This is not clear" The resulting activity patterns can be interpreted as the differences between melodies that underlie successful decoding". Rephrase, or just delete since the line before seems adequate.

In "The same areas represented the melodies after the third sound (p ≤ .021; Fig. 3c)" , make clear whether you mean as for the first or second sound.

Elaborate on this statement" Furthermore, the detection of deep sources has become increasingly common, including the basal nuclei, the medial temporal lobe and even the cerebellum (43-48)". Why? Better methods?

---

## [Editor Report · Decision Letter 2]

3 Sep 2024

Dear David,

Thank you for your patience while we considered your revised manuscript "Decoding reveals the neural representation of perceived and imagined musical sounds" for publication as a Research Article at PLOS Biology. This revised version of your manuscript has been evaluated by the PLOS Biology editors and the Academic Editor.

Based on our Academic Editor's assessment of your revision, we are likely to accept this manuscript for publication, provided you satisfactorily address the following data and other policy-related requests:

* Please add the links to the funding agencies in the Financial Disclosure statement in the manuscript details.

* Please state in the methods section whether the participants provided written or oral consent.

* DATA POLICY:

You may be aware of the PLOS Data Policy, which requires that all data be made available without restriction: http://journals.plos.org/plosbiology/s/data-availability . For more information, please also see this editorial: http://dx.doi.org/10.1371/journal.pbio.1001797

Regardless of the method selected, please ensure that you provide the individual numerical values that underlie the summary data displayed in the following figure panels as they are essential for readers to assess your analysis and to reproduce it: Figures 1C, A6 and A7.

* CODE POLICY

We expect to receive your revised manuscript within two weeks. 

*Published Peer Review History*

*Press*

Sincerely,

Christian

Christian Schnell, PhD

Senior Editor

cschnell@plos.org

PLOS Biology

---

## [Editor Report · Decision Letter 3]

20 Sep 2024

Dear David,

Thank you for the submission of your revised Research Article "Decoding reveals the neural representation of perceived and imagined musical sounds" for publication in PLOS Biology. On behalf of my colleagues and the Academic Editor, Manuel Malmierca, I am pleased to say that we can in principle accept your manuscript for publication, provided you address any remaining formatting and reporting issues. These will be detailed in an email you should receive within 2-3 business days from our colleagues in the journal operations team; no action is required from you until then. Please note that we will not be able to formally accept your manuscript and schedule it for publication until you have completed any requested changes.

Please take a minute to log into Editorial Manager at http://www.editorialmanager.com/pbiology/ , click the "Update My Information" link at the top of the page, and update your user information to ensure an efficient production process.

PRESS

We frequently collaborate with press offices. If your institution or institutions have a press office, please notify them about your upcoming paper at this point, to enable them to help maximise its impact. If the press office is planning to promote your findings, we would be grateful if they could coordinate with biologypress@plos.org . If you have previously opted in to the early version process, we ask that you notify us immediately of any press plans so that we may opt out on your behalf.

We also ask that you take this opportunity to read our Embargo Policy regarding the discussion, promotion and media coverage of work that is yet to be published by PLOS. As your manuscript is not yet published, it is bound by the conditions of our Embargo Policy. Please be aware that this policy is in place both to ensure that any press coverage of your article is fully substantiated and to provide a direct link between such coverage and the published work. For full details of our Embargo Policy, please visit http://www.plos.org/about/media-inquiries/embargo-policy/ .

Sincerely, 

Christian

Christian Schnell, PhD

Senior Editor

PLOS Biology

cschnell@plos.org